# A Systematic Review of EMG Applications for the Characterization of Forearm and Hand Muscle Activity during Activities of Daily Living: Results, Challenges, and Open Issues

**DOI:** 10.3390/s21093035

**Published:** 2021-04-26

**Authors:** Néstor J. Jarque-Bou, Joaquín L. Sancho-Bru, Margarita Vergara

**Affiliations:** Department of Mechanical Engineering and Construction, Universitat Jaume I, E12071 Castellón, Spain; sancho@uji.es (J.L.S.-B.); vergara@uji.es (M.V.)

**Keywords:** ADL, EMG, forearm muscles, muscles role, synergies, muscle coordination

## Abstract

The role of the hand is crucial for the performance of activities of daily living, thereby ensuring a full and autonomous life. Its motion is controlled by a complex musculoskeletal system of approximately 38 muscles. Therefore, measuring and interpreting the muscle activation signals that drive hand motion is of great importance in many scientific domains, such as neuroscience, rehabilitation, physiotherapy, robotics, prosthetics, and biomechanics. Electromyography (EMG) can be used to carry out the neuromuscular characterization, but it is cumbersome because of the complexity of the musculoskeletal system of the forearm and hand. This paper reviews the main studies in which EMG has been applied to characterize the muscle activity of the forearm and hand during activities of daily living, with special attention to muscle synergies, which are thought to be used by the nervous system to simplify the control of the numerous muscles by actuating them in task-relevant subgroups. The state of the art of the current results are presented, which may help to guide and foster progress in many scientific domains. Furthermore, the most important challenges and open issues are identified in order to achieve a better understanding of human hand behavior, improve rehabilitation protocols, more intuitive control of prostheses, and more realistic biomechanical models.

## 1. Introduction

The ability to carry out activities of daily living (ADL) is critical to ensure a full and autonomous life [1], and has been established by the WHO as the main factor for classifying the degree of disability [2]. ADLs refers to those elementary tasks that allow anyone to function with minimal autonomy and independence, including any daily activity that we perform for self-care, work, housework, and leisure. The ability of the hands to grasp and manipulate is fundamental in the performance of ADL [3] but also for working life [4]. This ability is achieved thanks to a complex musculoskeletal system, with 25 degrees of freedom that are controlled by approximately 38 muscles located in the forearm and hand [5]. These muscles can be divided into two groups: extrinsic and intrinsic muscles. The extrinsic muscles are located in the anterior and posterior compartments of the forearm whereas intrinsic muscles are located within the hand itself. Broadly, the extrinsic muscles are considered to control crude movements of the hand and produce forceful grip, while the intrinsic muscles would be responsible for the fine motor functions of the hand [6,7,8]. However, both the specific role of the different muscles in ADL and how the Central Nervous System dares with this redundant and complex muscular system are still unknown [9]. This information is essential for determining the impact on functionality when a given muscle is compromised because of an accident or pathology.

The measurement and interpretation of the hand kinematics and the associated muscle activation signals is complex but of great importance to deepen the knowledge of the role of the muscles in ADL [8]. This knowledge is not only important to rate disability but also to improve rehabilitation processes [10,11] or to help in decision-making during surgical planning, among others [12]. Another important field of application is in the control of hand prostheses [13,14,15,16]. Myoelectric hand prostheses use the electrical action potential of the residual muscles in the limb emitted during muscular contractions. These emissions are measured on the skin surface, picked up by electrodes, and are amplified to be used as control signals for the functional elements of the prosthesis. Therefore, deepening the knowledge of the role of the forearm muscles in ADL may help in the selection of the muscles to control this type of prostheses.

Electromyography (EMG) emerged as a diagnostic procedure to assess the health of muscles and the nerve cells that control them (motor neurons). The electrodes receive the electrical signals transmitted by the motor neurons that cause muscle contraction. However, these EMG signals acquired from muscles require advanced methods for their detection, decomposition, processing, and classification [17,18,19] that a specialist interprets. There are two basic types of electrodes to acquire these signals: surface and intramuscular electrodes. Surface electrodes are placed on the skin directly over the muscles, recording the signal from all the fibers under the two electrodes. Intramuscular electrodes can be indwelling (also known as needle) or fine wire electrodes (Fw-EMG), and they are inserted through the skin directly into the muscle [20], thus recording the signals from only few fibers. The general advantage of surface electrodes is that they are non-invasive and easy to apply. Their use, however, is limited to superficial muscles that are large enough to support electrode mounting on the skin surface, and crosstalk is particularly problematic for smaller muscles within a complex mechanical arrangement, such as the forearm [21]. Indwelling electrodes need significantly more training for their proficient use in comparison to surface electrodes. Although they are ideal for recording the activity of deep muscles, correct placement requires a detailed knowledge of musculoskeletal anatomy. Furthermore, the invasiveness of inserting a needle into the muscles, as well as the associated pain, is the major disadvantage of intramuscular electrodes [20].

EMG has been incorporated as a diagnostic technique for the detection of pathologies that affect nervous and muscular structures, and for the spatial location of the origin of the injury. Examination with needle EMG allows motor unit action potentials (MUAPs) to be evaluated. High density EMG grids allow also the identification of the MUAPs, but only for those motor units whose innervation zone was close to the surface of the muscle [22]. The morphology (duration, amplitude, and number of phases) and recruitment pattern of MUAPs are the key element for diagnosing pathologies using needle EMG. MUAPs are analyzed per muscle and the results compared with those normally expected for that particular muscle. For this reason, due to the need for a normal pattern, evaluation of MUAPs is not useful to gain deeper knowledge of the role of the muscles in ADL. Parameters, such as time-domain, time-frequency domain, or intensity of muscle activation, could be more useful for studying tasks. EMG data for these purposes are commonly normalized to a reference value to avoid variability arising from electrode placement, participants, or even the day of the experiment. The most popular method is to normalize EMG data to the maximum voluntary contraction (MVC) of the muscle of interest [23], which, besides making data comparable, also informs about how active muscles are relative to their maximum capabilities. Surface EMG is applied in many fields, such as motor control of human movement, myoelectric control of prosthetic and orthotic devices, and rehabilitation [24,25,26,27]. Some studies have performed EMG analyses for intrinsic and/or extrinsic hand muscles in specific situations: while grasping objects [28,29,30,31,32,33,34,35,36,37,38,39,40], during working postures [41,42,43,44,45,46,47,48,49], and for the design and improvement of sports equipment, as well as for the study of the role of muscles in sports performance [50,51,52,53,54,55].

The concept of synergy has been used in the field of control of myoelectric hand prostheses in an attempt to simplify the study of the complex kinematics and muscular action of the hand [56,57,58,59]. There are some studies describing muscle patterns or muscular synergies during some postures [60], grasps [61,62], or hand movements [60,63], and during particular actions [64,65,66,67,68]. In that research, different activation patterns have been obtained, revealing coordination between certain intrinsic and extrinsic hand muscles. Thus, EMG patterns have been studied as a way to control signals [69]. However, the usability of myoelectric prostheses is still challenged because of issues, such as the effect of electrode location or changes in EMG patterns over time, which can lead to long training processes [70]. A small number of studies have investigated the existence of hand muscle synergies in ADLs, which could help in the selection of muscles to control myoelectric prostheses.

In this work, a review of the studies in which EMG has been used to record the muscular activity of hand and forearm muscles during ADL is presented, which may help to identify the role of these muscles in ADLs. In addition, studies examining EMG patterns or muscular synergies between the muscles of the hand and forearm are also presented, in order to simplify the study of the muscular action of the hand. The contents are organized in two sections: muscle activation in different activities and hand muscle synergies (dimensional reduction of EMG).

## 2. Materials and Methods

The literature review consisted in examining research studies that recorded EMG of hand and forearm muscles regarding the activation of these muscles in ADLs, and the dimensional reduction of the muscular action of the hand. A systematic literature search was conducted in the Scopus and PubMed databases until March 2021. Figure 1 shows the flowchart followed. The search was restricted to papers published in English and containing the terms (“Electromyography” and “muscles”) and (“thumb” or “finger” or “hand” or “forearm”) in the title, abstract, or keywords. Then, a refined search was conducted including different keywords in the title, abstract, or keywords (see Figure 1). Finally, a manual screening was carried out to remove duplicates and reject non-relevant articles.

After the manual screening, 21 articles related to muscle activity during different ADLs were selected, and 21 articles related to dimensional reduction of EMG of the hand were identified. Altogether, 42 articles (including two reviews) were selected for inclusion in the current review. The articles selected are discussed in the following sections.

## 3. Results

### 3.1. Muscle Activation in Different Activities

This section includes a review of studies that have characterized hand and forearm muscle activity while performing specific activities, such as grasps, ADL, work activities, and sports. Table 1 summarizes the most relevant information of the 21 papers related to muscle activity during different ADL found in the literature.

Several studies have analyzed the activation of hand and forearm muscles (both extrinsic and intrinsic) during certain types of grasps. Regarding extrinsic muscles, they found that in power grasps both the flexor and extensor groups of muscles (extrinsic muscles) were activated, although the extensor part underwent greater fatigue [47]. As regards the intrinsic muscles, they found that during precision grasps intrinsic muscles play a major role in finely graded force generation, since fine movements require less stabilization and counterforce to the long flexor action [36].

Many studies have focused specifically on the thumb muscle activation through EMG while performing different grasps [30,37,38,39,40,71], especially during the opposition movement, due to its great importance in precision grasps. In general, these studies found a need for a cooperation of thumb muscles to accomplish the tasks performed [39], with the exception of the Extensor Policis Longus and Flexor Policis Longus (EPL, FPL), which could be activated separately from the other muscles [40]. Another study [30] explored and demonstrated the importance of the opposition of the thumb during stable and unstable lateral grasps. They observed that instability affects some thumb muscles with greater activation of Abductor Policis Longus (APL) and EPL in the unstable tasks. Similarly, Kaufman et al. [71] recorded the EMG activity of 7 thumb muscles and their contribution at the carpometacarpal (CMC) joint during voluntary isometric contractions. They found that:Thumb CMC flexion is supported by Flexor Policis Brevis (FPB), Abductor Policis Brevis (APB), thumb Adductor (ADD), and FPL.CMC extension by APB, APL, and EPL; CMC abduction by FPB, APB, APL, and EPL; and CMC adduction by FPB, APB, EPL, and FPL.The Opponens Policis (OPP) was active in all motions.

However, the studies in the literature all focused on small sets of very controlled and simple activities (a few grasps or simple finger movements). Additionally, they were limited to very specific muscles, or specific fingers or joints, especially for the role of the thumb.

EMG has also been used to study the effect of different kinds of work activities on the forearm muscles, evaluating the influence of different factors on fatigue during repetitive tasks [41,43,49], such as typing, keying, writing, reading and mousing, and on pulling and pushing tasks [45,46]. These studies have focused on evaluating and comparing different forearm and hand positions. Nevertheless, the relationship between force production and EMG is not well understood, and there are factors that influence the forces generated and therefore prevent the direct quantification of muscle force from EMG signals. These factors include variations in the location of the recording electrodes, crosstalk, the involvement of synergistic muscles, properties of muscles, tendons, ligaments, etc. Consequently, the EMG-force relationship differs for each muscle and for each situation [72]. Many studies in the literature have focused on examining muscle activation of both intrinsic and extrinsic muscles in writing activities [31,32,35]. One of these studies [35] compared two different typewriting tasks and the results suggested that the major function of the Extensor Carpi Radialis (ECR) muscle as a stabilizer of the wrist joint is maintained during handwriting. It is also suggested that the increased use of extrinsic muscles could result in a diminished role of intrinsic hand muscles. In that research, the authors showed that EMG of hand and arm muscles may be converted into handwriting patterns. However, the results of those studies were focused on specific activities and in many cases with a low number of subjects and activities, lacking representativeness in ADL.

EMG has also been used in the design and improvement of sports equipment, as well as for the study of the role of muscles in sports performance. Some researchers have studied the regions that are activated, thereby making the main movement possible, in sports, such as golf [50] and tennis [51,52,53,54]. Other studies have focused on examining the effect of different features of sports equipment, such as the size of the handle of rackets [53] or of a golf club [55].

Some authors [50,52] have observed that there is considerable diversity in the protocol design used for sEMG recording. For example, most of the studies did not specify the electrode placement, so it is not clear which locations were used to acquire the EMG data, thus making it difficult to compare values. A recent study [73] recorded sEMG activity from 30 spots distributed over the skin of the whole forearm of six subjects while performing 21 representative ADL from the Sollerman Hand Function Test (SHFT). As a result, they proposed that the number of sEMG sensors could be reduced from 30 to 7 without losing any relevant information, using them as representative spots of the muscular activity of the forearm in ADL.

There are few EMG analyses of upper extremities that examine muscle function during daily tasks, and they use little variability and a limited number of tasks (no more than 10) [74,75,76]. A wide variety of clinical tests (such as the Jebsen Taylor Hand Function (JTHF) test [77], Chedoke Arm and Hand Activity Inventory (CAHAI) [78], or SHFT [79]) are often used to evaluate and track functional recovery of the upper extremity simulating ADL. In these cases, EMG recordings may provide a window into the central nervous system to evaluate muscle recruitment and coordination. In this sense, Peters and collaborators [75] evaluated the recruitment and coordination between several upper-limb muscles during some of these clinical tests (JTHF, CAHAI, and Block and box test (BBT)). Specifically, they recorded sEMG from eight upper-extremity muscles (Anterior and Posterior Deltoid (AD and PD), Biceps Brachii (BB), Triceps lateral head (TriB), Brachioradialis (Br), ECR, Flexor Carpi Ulnaris (FCU), and Extensor Digitorum (EDC)), and evaluated which muscles were used to execute each task and whether activation and co-contraction levels were similar across tasks. As results, they found that co-contraction levels were similar across tests and EDC was found to have the greatest activation levels across all tasks, thereby denoting its importance for common tasks. However, this study has several limitations: they evaluated a small set of forearm muscles (only four forearm and finger muscles).

Summing up, most research found in the literature presents gaps that require further investigation, as, in many cases, the studies are focused on small sets of very controlled and simple activities. Few of them characterize the EMG activity of all the hand muscles while performing representative actions, either by carrying out all possible grasp types required in ADL or by performing a representative and conveniently standardized set of ADL. Furthermore, the lack of a methodology and a standardized protocol hinders the comparison of EMG results between tasks and subjects. Indeed, at the hand level there are few specific recommendations to help in this decision, although results from a recent study [73] could assist in this task, as 7 specific spots were identified as being representative of the muscular activity of the forearm in ADL. In addition, more studies are required to improve the knowledge about the relationship between force production and EMG.

**Table 1 sensors-21-03035-t001:** Summary of the studies included in the systematic literature review (I). Relevant information contains subjects, type of EMG used, and muscles recorded in the studies.

Study	Relevant Information	Description of the Task	Observations about Role of Muscles
Cooney et al. [38]	8 healthy subjects Fw-EMG Extrinsic muscles: FPL, APL, EPL, EPB Intrinsic muscles: ADD, APB, OPP	Isometric F/E and Abd/Ad thumb movements, pinch and power grasps	Extensor muscles (EPL, EPB, and APL) were primary and contributed nearly equally to the extension. In flexion, only the FPL was primary. ADD and APB are primary in adduction but the EPL (adduction) and OPP (abduction) contribute significantly. Three muscles appear to be primary in pinch and power grasp: the ADD, OPP, and FPL.
Kilbreath and Gandevia [80]	7 healthy subjects needle EMG Extrinsic muscles: FPL, FDP Intrinsic muscles: None	Dynamic movements: Lift a weight by flexion of one digit	There is a coactivation of other deep digital flexor muscles and this coactivation increases when the digit flexes at a greater velocity or through a larger angle.
Maier and Hepp-Reymond [36]	6 healthy subjects needle EMG Extrinsic muscles FDP, FDS, APB, FPL, EPL EPB, APL, EDC, EI Intrinsic muscles: FDI, FPI, FPB, FLUM ADD, OPP	Isometric forces with thumb and index finger	The intrinsic muscles (FDI, FPI, and FLUM) and the long flexors (FDP, FDS) of the index finger, as well as two intrinsic muscles of the thumb (ADD FPB), increase their activity according to the load. The other thenar muscles (OPP, APB) and the extrinsic muscles of the thumb (FPL, EPL, EPB, and APL) become active only at higher loads and may serve to stabilize joints. The long extensors of the index finger (EDC, EI) were classified as antagonistic, and only act to balance the applied load and maintain joint equilibrium. The intrinsic muscles play a major role in finely graded force generation since less stabilization and counterforce to the long flexor action are needed, thus releasing the intrinsic muscles for precise force regulation.
Hägg and Milerad [47]	9 healthy subjects sEMG Extrinsic muscles: FCU, FDS, ECR Longus, ECR Brevis, EDC Intrinsic muscles: None	Simulations of grips in industrial work	Fatigue effects are generally larger on the extensor side although none of the regimes studied were acceptable from point of view of EMG fatigue.
Chow et al. [51]	7 healthy and skilled tennis subjects sEMG Extrinsic muscles: FCR, ECR Intrinsic muscles: None	Tennis volley	In general, the ECR was more active than the FCR during the volley, suggesting the presence of wrist extension and abduction.
Kaufman et al. [71]	5 healthy subjects Fw-EMG Extrinsic muscles: APL, FPL, EPL Intrinsic muscles: APB, OPP, FPB, ADD	Isometric thumb motions in F/E and Abd/Ad	The FPB was most active in the range from flexion to abduction with about 50% activity or less in extension and adduction. The OPP displayed activity in all directions of motion. The APB displayed maximal activity in abduction and abduction-flexion directions. The ADD was active during flexion. The APL was most active in abduction and/or extension. The EPL showed the highest activity during extension in combination with abduction/adduction functions. The FPL was the most active in flexion and/or adduction functions.
Johanson et al. [30]	7 healthy subjects Fw-EMG Extrinsic muscles: FPL, EPL, EPB, APL Intrinsic muscles: APB, ADD, FDI, FPB OPP	Key and opposition pinch postures between stable and unstable tasks	Activation patterns are different between key and opposition pinch posture and between stable and unstable pinch tasks. APB and EPL muscles are necessary to accurately direct thumb-tip forces in a functional pinch, not just to position the thumb, independently of pinch force magnitude. In all unstable conditions, APB and EPL were among the most activated muscles and could provide the task with directional accuracy.
Van Galen et al. [41]	20 subjects sEMG Extrinsic muscles: BB, TriB, FCU, ECR Intrinsic muscles: None	Fast movements with an electronic pen along the surface of a digitizer	For the forearm muscle movers, our findings show that the participants reacted with a substantial increase in static muscle activity, i.e., of antagonistic activation. For the wrist stabilization musculature, however, the effect was reversed.
Danion and Galléa [37]	7 healthy subjects sEMG Extrinsic muscles: EPL Intrinsic muscles: FPB	Constant force matching task during precision grasp	Muscle co-contraction is not a critical factor for force steadiness during a precision grasp task. Muscle co-contraction and grip force steadiness depend on grip force magnitude, but grip force steadiness does not depend on muscle co-contraction.
Hatch et al. [53]	16 healthy tennis players Fw-EMG Extrinsic muscles: EDC, ECR Longus, ECR Brevis, PT, FCR Intrinsic muscles: None	Back-hand tennis stroke	There was a progressive increase in ECRL and FCR activity from early acceleration through ball impact. There was a progressive increase in EDC activity through the early and late acceleration phases. At ball impact, all muscle activity tended to peak, and then gradually decreased in the early follow-through phase.
Ertan [67]	20 healthy subjects sEMG Extrinsic muscles: EDC, FDS Intrinsic muscles: None	Dynamic archery shooting	Elite archers relax their finger flexors so as not to grip the bow-handle, and contract the extensors to avoid holding/gripping the handle throughout the whole shot. The main difference between elite and beginner archers was that the elite archers had a greater activation of the EDC.
Linderman et al. [32]	6 healthy subjects sEMG Extrinsic muscles: FCR, EDC, ECU, ECR Intrinsic muscles: OPP, APB, FDI	Writing numeric characters	It is feasible to recreate handwriting solely from EMG signals thanks to the existence of muscle patterns during writing.
Di Dominizio and Keir [46]	12 healthy subjects sEMG Extrinsic muscles: FCR, FCU, FDS, ECR, ECU, EDC, AD, PD Intrinsic muscles: None	Grips with pull and push tasks	Flexor muscle activity tended to be lower when performing push with grip tasks and pull with grip tasks than extensor muscle activity. The highest wrist and finger extensor (ECR, ECU, and EDC) activity was elicited when performing grip tasks in a pronated posture. ECU was found to be the most sensitive to postural changes.
Szeto and Lin [49]	17 healthy subjects and 9 symptomatic office workers sEMG Extrinsic muscles: ECR, FCU, ECU, FCR Intrinsic muscles: None	Performing mouse-clicking tasks under different speed and precision conditions	Higher EMG amplitudes in the Control Group over Case Group mostly in the ECU and ECR muscles and more so in the more stressful condition. ECR muscle recorded significant group differences in both precision and speed condition analyses, and FCU in speed condition analysis.
Marta et al. [50]	Review paper about amateur and professional golfers sEMG/Fw-EMG Extrinsic muscles: ECR Brevis, PT, FCR, FCU, Intrinsic muscles: None	Different phases of the golf swing	Higher peak activity in the leading PT during the acceleration phase and just after the impact in professional golfers compared to amateur players who showed peak activation in the ECRB. This study also reported considerably higher levels of activity in the ECRB in amateurs during all swing phases. Some studies did not specify the electrode placement, so it is not clear which locations were used to acquire the EMG data, thus making it difficult to compare values.
Almeida et al. [35]	24 healthy subjects sEMG Extrinsic muscles: Tr, BB, ECR Brevis, FDS Intrinsic muscles: None	Writing a word five times	The major function of the ECRB muscle as a stabilizer of the wrist joint is maintained during handwriting tasks and the increased use of extrinsic muscles could result in a diminished role of intrinsic hand muscles during handwriting.
Birdwell et al. [40]	7 healthy subjects Fw-EMG Extrinsic muscles: APL, EPB, EPL EDC, FDP, FDS, FPL Intrinsic muscles: None	Activating each specific muscle during 3 s of MVC	Only two extrinsic thumb muscles, EPL and FPL, were capable of sustaining individual activations from the other thumb muscles. Activation of EPB elicited coactivity levels from EPL and APL.
Park [31]	36 healthy subjects sEMG Extrinsic muscles: FCU, FCR, ECU, ECR, Upper Tr Intrinsic muscles: None	Writing subtests of the JHFT	ECU was the most active muscle during writing in both left- and right-handers.
Alizadehkhaiyat and Frostick [52]	Review paper: 198 healthy (controls) 18 Lateral-epicondylitis patients sEMG/Fw-EMG Extrinsic muscles: FCU, FCR, ECR, ECU, PT Intrinsic muscles: None	Different tennis strokes	Increase in the activity of wrist extensors including ECR Brevis and ECR Longus in multiple phases of forehand, serve, and backhand strokes with the activity of wrist flexors remaining fairly constant. Higher EMG activity of ECR during repetitive pre- and post-impact in the presence of unchanged FCR activity has been suggested as predisposing players to injury or delaying the recovery process. Finally, an earlier, longer, and greater activation of ECR Brevis during backhand volleys at combined conditions of velocity and racket-head impact locations has been reported in LE patients compared to non-injured players. There was considerable diversity in the protocol design used for EMG recording.
Kerkhof et al. [39]	10 healthy subjects Fw-EMG Extrinsic muscles: FPL, EPB, EPL, APL Intrinsic muscles: ADD, APB, FPB OPP	Isometric contractions in a lateral key pinch, a power grasp, and a jar twist task	Extrinsic thumb muscles were significantly more active than intrinsic muscles in all tasks. The thumb muscles display a high variability in muscle activity during functional tasks of daily life. To produce a substantial amount of force, a co-contraction between the intrinsic and extrinsic thumb muscles is necessary.
Peters et al. [75]	20 healthy subjects sEMG Extrinsic muscles: AD, PD, BB, TriB, BB, ECR Longus, FCU, EDC Intrinsic muscles: None	Clinical tests of upper extremity function	Minimal muscle force is required to perform these tests. Co-contraction levels were similar across tests. EDC has the greatest activation levels across all tasks. The results suggest that healthy participants used different strategies to execute the tests.
Jarque-Bou et al. [73]	6 healthy subjects sEMG Extrinsic muscles: Muscles recorded from all the forearm Intrinsic muscles: None	21 ADL selected and adapted from SHFT	The EMG sensors needed to record relevant information about forearm muscle activity could be reduced to 7. The signals from these seven spots would be related to seven different movements: wrist flexion and ulnar deviation wrist flexion and radial deviation digit flexion thumb extension and abduction/adduction finger extension wrist extension and ulnar deviation wrist extension and radial deviation
Jarque-Bou et al. [81]	22 healthy subjects sEMG Extrinsic muscles: Seven spots representative of all available muscle activity of the whole forearm Intrinsic muscles: None	26 representative ADL	Minimal muscle force is required to perform ADL. Greater activity is shown during reaching (to place the hand to grasp) than during manipulation. Finger and wrist extensors were the most active muscles while performing ADL. Muscle activity presented some variability among subjects, highlighting the different possibilities that each subject may have to carry out the same activities.

Abbreviations: FLUM—First Lumbrical; FDI—First Dorsal Interosseous; FPI—First Palmar Interosseous; ECU—Extensor Carpi Ulnaris; PT—Pronator Teres; FCR—Flexor Carpi Radialis; FDS—Flexor Digitorum Superficialis; FDP—Flexor digitorum Profundus.

### 3.2. Hand Muscle Synergies

This section includes a review of studies that have characterized hand and forearm muscle activity by studying EMG patterns or muscular synergies between the muscles of the hand and forearm in order to simplify the study of muscular action of the hand. Table 2 summarizes the most relevant information in the 21 papers related to hand muscle synergies that were selected.

The human hand has a complex biomechanical structure, controlled by a neural structure that is still not completely understood. In the analysis of the biomechanical and behavioral aspects of the hand, one of the most striking is the high redundancy of its structure, seemingly having many more muscles than are actually required [56]. Synergies are thought to be used by the nervous system to simplify the control of these numerous muscles by actuating them in task-relevant subgroups. There are studies both for and against muscle synergies [82]. Many researchers seek to detect and describe such simplifying functional muscle groups and how to interpret them in order to reveal the underlying control strategy used by the brain to coordinate muscles [56,60]. Others point out the importance of the ability of the brain to break and dissolve such patterns of neural synchrony. This might happen to enable flexible and individuated control of hand muscles [83,84,85], thus indicating that muscles are recruited flexibly in accordance with their mechanical action, rather than in fixed groupings. In both cases, EMG of hand muscle activity has been extensively used to infer the control strategies underlying the complex coordination of muscle activity within and across digits and as a tool to study the spatial and temporal coordination of multiple muscles. In fact, this technique has been used to examine the organization of these muscle synergies in healthy and neurologically impaired individuals [83,86]. In addition, several studies have examined the covariations in EMG amplitudes across muscle pairs [36] and among multiple muscles [87,88] related to hand function.

Previous studies reinforce the idea of hand muscle synergies, and their results concerning the features of hand muscle synergies can be summarized as: muscle synergy occurs primarily across muscles with similar mechanical features [89]; the coactivity of some muscles is a way to adapt the limb to different environmental conditions [90,91]; and the whole set of hand and forearm muscles may be approximated with relatively few adequately scaled and synchronized muscle synergies [60,92,93,94,95,96].

The basic approach of these studies involves four steps:(1).Measuring sEMG from a large number of muscles during a complex behavior (or more than one behavior). Note that surface electrodes are the most widely used method, since they are non-invasive and a high number of muscles on the forearm need to be recorded.(2).Using a computational analysis, such as non-negative matrix factorization, to identify a set of synergies from the sEMG recorded. Different factorization methods have been used to extract muscle synergies from muscle activation patterns during dynamic tasks. The two most commonly used factorization methods reported in the literature are non-negative matrix factorization and principal component analysis [97]. Similar results are obtained in both cases, but the non-negative matrix factorization method is the most suitable when recording a high number of channels [98].(3).Evaluating whether the sEMG observed can be well described as the combination of these synergies.(4).Relating the muscle synergies identified to task-relevant variables [82].

As a result of the application of this procedure, two different types of synergies are described in the literature: synchronous synergies and time-varying synergies. A synchronous synergy is a vector of weighting coefficients that specify the relative involvement (strength of membership) of each muscle in the group. In contrast, a time-varying synergy is a collection of EMG bursts in various muscles.

Several studies describe muscle patterns or muscular synergies during certain specific postures or grasps [99] or during the whole-hand grasping performance [60,62]. Some synergies during specific tasks are also described, such as during finger spelling [60,95], or the preparatory muscle activation response when a fall occurs [100]. Weiss and Flanders [60] recorded the EMG activity of 6 hand and forearm muscles (APB, FPB, FDI, EDC, ADM, and FDS) in four subjects while they held the hand statically, shaping around 26 grasped objects and forming the 26 letter shapes of a manual alphabet. They found that a single muscle can be a member of more than one muscle synergy [60,101]. Klein Breteler et al. [95] expanded the synergy analysis from static synergies to time-varying synergies in order to explore the timing of muscle activations during finger spelling using a manual alphabet. They recorded FDI, APB, FPB, ADM, FDS, and EDC and concluded that four time-varying synergies could account for 80% of the temporal EMG patterns observed, with the first two synergies accounting for about 60%. In addition, they showed that the first component displayed a consistent pattern, the first and second component waveforms showed similarities across subjects, and higher order components were far more variable across subjects. The first component was a pattern where the EDC and the thumb muscles (APB and FPB) were active early on, and the other muscles were active later. Recently, Scano and collaborators [62] extracted muscle synergies from 20 hand grasps with an array of 8 equally spaced electrodes on the forearm, two electrodes on finger flexors and extensors, and another two on BB and TriB. The synergies they found were characterized by two temporal activation patterns: a strong coactivation corresponding to the grasp/hold phase, and two minor coactivating patterns related to hand opening (visible in the pre-shaping and release phase).

Synergistic finger patterns have also been described during dynamic free movements of the wrist and single fingers [102,103,104]. Tanzarella et al. [104] performed isometric contractions with each finger and with three combinations of fingers in opposition with the thumb. They observed a low dimensional control of motor neurons across multiple intrinsic and extrinsic muscles. Gazzoni et al. [102] identified distinct areas of sEMG activity on the forearm for different fingers during hand and finger movements. In the same way, Hu et al. [103] revealed distinct activation patterns during individual finger extensions, especially for the index and middle fingers. Nevertheless, the detailed location of the recording electrodes was not reported in most of the studies, which makes comparison between subjects and activities difficult.

However, few studies have assessed muscle patterns during complex tasks in which manipulation is the most relevant phase, such as in ADL [61,68], or during particular actions, such as playing a piano [63], archery [65,67], catching a ball [64], or while performing a karate punch [66]. Winges et al. [63] studied the muscle activation patterns of 10 pianists, suggesting that amateur pianists use the same balance as professionals. Nevertheless, in other research [65,67], the authors found different patterns between elite and beginner archers, where the main difference was that the expert archers had a greater activation of the ED. In this sense, in a study on karate punches [66], expert and non-expert karatekas presented distinct kinematic and EMG patterns. Regarding a more complex activity, such as catching a ball [64], the authors recorded sEMG data from 16 shoulder and elbow muscles, but only one forearm muscle (Br), in six subjects. They found that the variation in the muscle patterns was captured by two time-varying muscle synergies, modulated in amplitude and shifted in time according to the height at which the ball arrived and the flight duration. The initial muscular response, captured by the first synergy, allowed the subject’s hand to reach the interception zone. The following component of the muscle pattern, captured by the second synergy, guided the hand to the interception. Zariffa et al. [61] characterized what muscle synergies were present while using different types of hand grips (gripping a block, a cylinder, a ball, a key, and rotating a disk 180 degrees) extracted from clinical tests. sEMG data was recorded from FDI, FCU, FCR, FDS, ECR, EDC, EIP, and the thenar eminence muscle group. Two main synergies were found: the first between EDC and EIP, and the second between FDS and FCU. However, they had some limitations due to crosstalk, the small number of muscles recorded, and the little variability of the ADLs chosen. Ricci et al. [68] recorded data from shoulder and elbow muscles along with FDS, FCU, ECRLB, and ECU while subjects poured water. In the transport phase, characterized by weight bearing, handgrip and displacement of the arm in space, a higher activity of almost all muscles was found. Furthermore, they found that ECR seems to play a key role in maintaining optimal wrist posture and function regardless of task demand. That stabilization could be provided by a delicate balance of co-contraction of forearm muscles to keep the hand in the proper posture to grasp or produce handgrip force [105]. However, few forearm muscles were measured and for very specific actions, and therefore further studies should be conducted to evaluate more forearm muscle patterns in a wide range of ADL. Likewise, hand kinematics should be recorded in order to relate muscular and kinematic hand synergies during representative ADL.

Moving on to the assessment of pathologies, as mentioned above, sEMG has recently been used for the evaluation of patients with neuromuscular disorders by using muscle synergies. Muscle synergies have been investigated in acute, subacute, and chronic stroke, showing abnormalities compared to healthy people [84,106,107,108,109], as well as in patients with dystonia [110] and sclerosis [111] or after spinal cord injury [61]. The results illustrate that muscle synergy patterns contain rich information in their spatial components and temporal profiles. Comparing pathological synergies of patients with the baseline synergy can reveal deficits in the underlying neuromuscular coordination and control. The analysis of task-specific muscle synergies should offer both researchers and clinicians new insights into the impairments in the neural organization of motor control. Yet, in these studies, a considerable diversity in the protocol design was used for sEMG recording, and it is not clear which locations were used to acquire the sEMG data, resulting in difficulty when it comes to comparing values.

Summarizing, EMG has been widely used to detect muscle patterns, although a small number of studies have investigated the muscular synergies in the hand in greater depth. There are some gaps that need to be studied in more detail. First, muscular synergies seem to be task-dependent, and a single muscle can be a member of more than one muscle synergy. In the literature, researchers have generally investigated the presence of synergies during some specific hand movements or grasps, but few studies have analyzed the different coordination and muscular patterns or synergies during the performance of a representative set of ADL. Second, little has been studied about kinematics and muscular synergies of the forearm and hand relationship. Consequently, little is known about the role of the muscles linked to the joint movement of the hand during ADL.

**Table 2 sensors-21-03035-t002:** Summary of the studies included in the systematic literature review (II). Relevant information contains subjects, type of EMG used, and muscles recorded in the studies.

Study	Relevant Information	Description of the Task	Observations about Muscles Role
Valero-Cuevas et al. [87]	8 healthy subjects Fw-EMG Extrinsic muscles: FDP, FDS, EI, EDC Intrinsic muscles: FLUM, FDI, FPI	Static force in five directions	CNS is implementing the predicted mechanically advantageous strategies, and scaling them down to produce less than maximal forces. Palmar force used flexors, extensors, and FDI. Dorsal force used all muscles. Distal force used all muscles except for extensors. Medial and lateral forces used all muscles including significant co-excitation of FDI.
Valero-Cuevas [88]	8 healthy subjects Fw-EMG Extrinsic muscles: FDP, FDS, EI, EDC Intrinsic muscles: FLUM, FDI, FPI	Different levels of fingertip forces while maintaining their forefinger in a static posture	Significant muscle coordination patterns similar to those previously reported for 100% of maximal fingertip forces were found for 50% of maximal voluntary force.
Weiss and Flanders [60]	4 healthy subjects sEMG Extrinsic muscles: EDC, FDS Intrinsic muscles: ADM, APB, FPB, FDI	Static postures for 26 objects and 26 letter shapes of a manual alphabet	Single muscles may be a member of more than one muscle synergy.
Klein Breteler et al. [95]	9 healthy subjects sEMG Extrinsic muscles: EDC, FDS Intrinsic muscles: FDI, APB, FPB, ADM	Finger spell words, presented on a computer screen	The first synergy represented the main temporal synergy, accounting for more of the EMG variance (up to 40%). This main synergy began with a burst in the EDC and a silent period in the flexors. There were then progressively later and shorter bursts in the APB, FPB, ADM, and, finally, the FDS.
Martelloni et al. [96]	6 healthy subjects sEMG Extrinsic muscles: TriB, Deltoid, Trapezius, FCR, ECR, BB Intrinsic muscles: None	Performing reach-to-grasp movements for different objects placed in different locations	Activation of proximal muscles can be statistically different for different grip types. Proximal and distal muscles are simultaneously controlled during reaching and grasping. Patterns of EMG activation in arm muscles can provide a reliable representation of motor behavior during reaching and grasping of different objects.
Valero-Cuevas et al. [85]	8 healthy subjects Fw-EMG Extrinsic muscles: EDC, EI, FDP, FDS Intrinsic muscles: FDI, FPI, FLUM	Vertical fingertip force vectors of prescribed constant or time-varying magnitudes	Evidence for preferential control of task-relevant parameters that strongly suggest the use of a neural control strategy compatible with the principle of minimal intervention. Only one synergy accounting for >40% of the variance with positive correlation among all muscles (coactivation). There was no reduction in dimensionality because each of the seven principal components explains a nontrivial amount of variance.
Marc H. Schieber et al. [83]	10 stroke subjects sEMG Extrinsic muscles: None Intrinsic muscles: APB, FDI, ADM	Cyclical F/E or Ab/Ad movements of each digit	FDI in the control hand was active only when the index finger was abducting. FDI in the affected hand was also active during movement of the thumb or the ring finger. These inappropriate contractions of FDI in the affected hand would cause the index finger to move when the subject attempted to move only the thumb or the ring finger. Muscle synergies of the stroke-affected arm were strikingly similar to those of the unaffected arm despite marked differences in motor performance between the arms. In subjects with severe motor impairment, there was far less resemblance between the synergies of the two arms.
VencesBrito et al. [66]	18 karatekas and 19 non-karatekas sEMG Extrinsic muscles: BB, Br, Deltoid, Pectoralis, PT, Infraspinatus Intrinsic muscles: None	Analysis of a karate punching movement (choku-zuki) on a fixed target	The two groups presented distinct EMG patterns. The first muscles to be activated were the agonists of the arm flexion and internal rotation. This was followed by an initial activation of the forearm flexor and pronator muscles. The forearm extensor muscle initiates its activity slightly later, followed by the second activation moment of forearm pronator muscle.
Cheung et al. [107]	31 stroke subjects sEMG Extrinsic muscles: infraspinatus; rhomboid major; Trapezius; pectoralis major; Deltoid; TriB; BB; brachialis, Br; supinator; PT Intrinsic muscles: None	Tasks and reaching movements with shoulder and forearm motions	Muscle synergies of the stroke-affected arm were strikingly similar to those of the unaffected arm despite marked differences in motor performance between the arms. In subjects with severe motor impairment, there was much less resemblance between the synergies of the two arms.
Zariffa et al. [61]	10 healthy subjects 6 Spinal cord-injured subjects sEMG Extrinsic muscles: FDS, FCR, FCU, ECR, EDC Intrinsic muscles: EI, FDI, Thenar eminence	7 functional tasks using grasp types relevant to ADLs	The synergies found were: (1) EDC and EIP, and (2) FDS and FCU. Many tasks involving finger extension tasks can be expected to recruit both EDC and EIP. The FDS and FCU synergy suggests that a wrist flexion was often used to position the hand during a grasping action, though this may be a product of the specific set of tasks employed in this study. The most common synergy in SCI subjects was FCR and ECR, which was also one of the average able-bodied synergies. FDI and Thenar eminence were common in both groups, possibly because of the need for independent fine thumb and index finger movements in many dextrous tasks. In subjects with SCI, similar synergies were observed, but in different proportions.
Burkhart and Andrews [100]	20 healthy subjects sEMG Extrinsic muscles: BB, Br, Anconaeus, FCR, ECR Intrinsic muscles: None	Impacts occurred to the hand from two heights	Individuals are capable of selecting an upper extremity posture that allows them to minimize the effects of an impact and the presence of a preparatory muscle activation response has been confirmed.
Castellini and Van Der Smagt [99]	6 healthy subjects sEMG Extrinsic muscles: Two bands surrounding the forearm Intrinsic muscles: None	Five static grasps: flat grasp, pinch grip, tripodal grip, small power grasp, and large power grasp	Three main synergies were found: uniform activation, activation of the dorsal muscles near the radius, and activation of the flexors near the radius.
Lee et al. [109]	4 healthy subjects 14 subjects with chronic hemiparesis sEMG Extrinsic muscles: FDS, EDC, FCR, FCU, ECR, ECU Intrinsic muscles: Thenar muscles, FDI, hypothenar muscles	Wrist F/E finger extension, lateral pinch, power grip, and tip pinch	The first synergy, containing mainly thenar and FDI activity, was largely active in the three grip tasks. The second synergy, consisting of EDC, ECR, and ECU, was heavily weighted during finger/wrist extension. The third synergy, involving coactivation of the wrist and finger muscles. The fourth synergy, with FCR, FCU, and EDC activity, was employed during wrist flexion. For stroke survivors, the composition of these modules was generally similar to those of subjects with no impairment.
Winges et al. [63]	10 healthy subjects sEMG Extrinsic muscles: FDS (2 portions), EDC Intrinsic muscles: ADM, APB, FPB, FDI	Piano dynamic movements: playing 14 selected pieces with the right hand at a uniform tempo	Phasic coactivation was evident between extensor and flexor muscles during piano playing. For the thumb sequence, PC1 first synergy was dominated by bursts of activity in the APB and the FPB with activity in the four-finger ED muscle. For the index finger sequence, the central burst of the first synergy included activity in two to three flexors of the index finger. Higher PC synergies were variable across subjects.
Hu et al. [103]	10 healthy subjects sEMG Extrinsic muscles: surface grid centered on the EDC Intrinsic muscles: None	Static and dynamic finger movements: To extend MCP joints individually	When the four fingers were extended simultaneously, the entire EDC was active. When individual fingers were extended separately, distinct regions of the EDC were selectively activated, with the index finger in the most distal region, the middle finger in the most proximal region, and the ring and little fingers in between. Index and middle fingers have a greater degree of individuation in comparison to the little and ring fingers.
Ricci et al. [68]	25 healthy subjects sEMG Extrinsic muscles: Trapezius, Serratus, Deltoid, Pectoralis, BB, TriB, FDS, FCU, ECR, ECU Intrinsic muscles: None	Pouring water task belonging to the Elui Functional Test of the Upper Extremity	In the reaching phase, the main movements observed were shoulder flexion and elbow and wrist extension, accompanied by significant higher activity of S, D, and TriB. The sequence of movements in this phase ended up with the subjects grasping the pitcher, which could be related to the late coactivation between ECU and FCU. Transport phase was mainly characterized by higher muscle activity of all muscles, except for Pectoralis. There were almost no significant differences in muscle activity within the release phase. ECR is a key muscle for wrist posture and function regardless of the task demand. Activation of FCU and ECRLB were identified as the main control strategy performed to maintain optimal grasping.
Roh et al. [108]	6 healthy subjects 16 post-stroke subjects sEMG Extrinsic muscles: Br, BB, TriB, Deltoid, and pectoralis Intrinsic muscles: None	Grasping the MACARM’s gimbaled handle	EMG spatial patterns were well explained by task-dependent combinations of only a few (typically 4) muscle synergies. Elbow-related synergies were conserved across stroke survivors, regardless of level of impairment. Alterations in the shoulder muscle synergies underlying isometric force generation appear prominently.
Hesam-Shariati et al. [110]	24 post-stroke subjects sEMG Extrinsic muscles: Trapezius, Deltoid medius, BB, ECR, FCR, FDI Intrinsic muscles: None	14-day program focused on the more- affected upper limb	The profile of coordinated muscle activation varied by the level of residual motor-function in chronic stroke. The number of synergies used increased (although not significantly) with therapy for patients with low and moderate motor-function. The distribution of muscle weightings within synergies changed as a consequence of therapy.
Lunardini et al. [111]	9 dystonia subjects 9 healthy subjects sEMG Extrinsic muscles: FCU, ECR, BB, TriB, Deltoid, Supraspinatus Intrinsic muscles: None	Writing task	Synergy analysis revealed no difference in the number of synergies between children with and without dystonia. Two synergies primarily involved upper limb distal muscles (distal synergies). Distal synergies were different depending on the task. The other two synergies mainly included proximal muscles (proximal synergies). Proximal synergies were very similar across groups and tasks: Synergy 3 involved shoulder flexors (D), while synergy 4 mainly comprised shoulder extensors (D and supraspinatus).
Pellegrino et al. [112]	11 healthy subjects 11 subjects with multiple sclerosis sEMG Extrinsic muscles: 15 upper limb muscles with only two forearm muscles (Br, PT) Intrinsic muscles: None	Reaching tasks: subjects grasped the handle	For both populations, the analysis identified three primary synergies that involved the distal muscles, another synergy that involved proximal muscles, and the last synergy included shoulder muscles. Muscle synergy analysis detected aspects related to muscle coordination that were not evident from the analysis of single muscle activity.
Scano et al. [62]	28 healthy subjects sEMG Extrinsic muscles: One band of 8 electrodes surrounding the forearm + BB, TriB, finger flexor and extensor Intrinsic muscles: None	Performance of 20 grasps	Ten spatial motor modules, properly elicited in time, are enough to describe the whole dataset with good accuracy, generalizing across subjects. The coactivating group composed of forearm electrodes is very often grouped together, especially in the hold phase. Two activation patterns are recognizable: a strong coactivation, often (but not always) corresponding to the grasp/hold phase, and two minor coactivating patterns in the pre-shaping and release phases that are often grouped in a single synergy. BB is activated during the reaching phase, thereby confirming that it is indeed an active reaching component that is active in the pre-shaping and release phase.

Abbreviations: ADM—Abductor Digiti Minimi; FLUM—First Lumbrical; FDI—First Dorsal Interosseous; FPI—First Palmar Interosseous; ECU—Extensor Carpi Ulnaris; PT—Pronator Teres; FCR—Flexor Carpi Radialis; FDS—Flexor Digitorum Superficialis; FDP—Flexor Digitorum Profundus.

## 4. Discussion

This literature review found 42 papers that matched the defined search criteria: 21 papers regarding the role played by hand and forearm muscles, and 21 dealing with hand muscle synergies.

First, studies focused on the role of specific small sets of forearm/hand muscles during some common tasks and grasps, sport activities, and working tasks were analyzed. Both extrinsic and intrinsic forearm muscles are required to accomplish these tasks, the extrinsic ones being responsible for the gross movements and the intrinsic ones in command of the fine movements, but they also complement each other. Some specific muscles show a high level of activation across all the tasks, such as EDC, while others seem to have a specific role, such as ECR as a wrist stabilizer. The tasks performed in ADL seem to require moderate levels of co-contraction of forearm muscles, needing the cooperation between different groups of muscles, this cooperation being non task-dependent [75]. Thumb muscles, such as EPL and EPB, are able to activate separately from the flexors and extensors of the other fingers, and their important role in grasps has been widely demonstrated in the literature.

Second, the muscle synergies reflecting the relationship between muscles provide information in two domains: co-contractions and timing of activation. Therefore, studying muscle synergies can help to reach a better interpretation of the role of the muscles during the execution of different movements/tasks. The idea is consistent with the concept that the central nervous system may embed a modular structure that relies on a limited number of synergies at hand level. Non-negative matrix factorization and principal component analysis methods are the most used and present similar results in terms of coordination patterns. However, the non-matrix factorization method is the most preferred when a high number of sEMG are recorded [98]. The studies reviewed have demonstrated that a small subset of synergies could be generalized across tasks, representing basic building blocks underlying natural human hand motions/actions. Therefore, muscle synergy analysis could also be useful for comparing different therapies and evaluating the function recovery of subjects regarding ADL performance. It has been hypothesized that patients’ functional deficit may be identified by regularly assessing their muscle synergy profile, which might be used to track the results of rehabilitation, and to adjust treatments [113]. Synergies have been suggested as being useful for clinicians to treat motor dysfunctions more effectively by organizing patients into subclasses and tailoring the treatment to each patient’s specific deficit [113].

However, some important gaps have also been identified, which should be addressed in further studies. One of the main gaps found in the literature is the considerable diversity in the protocol design used to record sEMG from forearm muscles. Most of the studies do not specify the electrode placement, so it is not clear which locations were used to acquire the sEMG data. This makes it difficult to compare values or may affect the crosstalk level, which will depend on the longitudinal level of the muscle where it has been placed. Therefore, it could be useful to define a method of placing the sEMG electrodes that is comparable between subjects and that considers all the muscles involved in wrist and hand movements. It has been seen that this could be achieved by identifying the most representative forearm areas for ADL performance in terms of EMG activity [73].

The second main gap concerns the lack of representativeness of the tasks used for the EMG characterization. Most studies found in the literature are focused on studying the role of specific muscles during simple tasks (hand postures or free finger movements), or during single activities (such as writing or typing), or during small sets of very controlled activities (a few grasps, sport movement, etc.). However, only a few studies have dealt with the analysis of the forearm and hand muscles during ADL, and none of them consider a wide representative set of ADL. Therefore, defining a selection of a limited set of representative tasks would improve the current methodology, given the wide variety of ADL that can be performed by humans. Furthermore, standardization of the tasks would allow for comparison between subjects and sessions (important for tracking function recovery). The use of standardized tasks is especially important considering that each different individual may perform the same activity using several different strategies. Standardization would help in the comparison of muscular patterns and the identification of different strategies, by distinguishing between the different task phases [114].

In addition, to go further into synergies, simultaneous measurement of hand kinematics is not usually performed, and, when it is measured, it is used only to segment the different phases of the movement. Therefore, linked EMG-kinematic datasets, at the hand level, are very limited [81]. Such synchronized datasets are needed if we want to analyze how hand movements are produced and controlled. This could be helpful in some fields, like rehabilitation (to help choose the most suitable approaches) or prosthetics (to find a more reliable and natural control of hand prosthetics).

The review performed provides a basis of knowledge about the role of hand/forearm muscles, but the lack of a clear methodology introduces some limitations. These methodological inconsistencies add additional difficulty for an effective interpretation of findings and to draw any decisive conclusions.

## Figures and Tables

**Figure 1 sensors-21-03035-f001:**
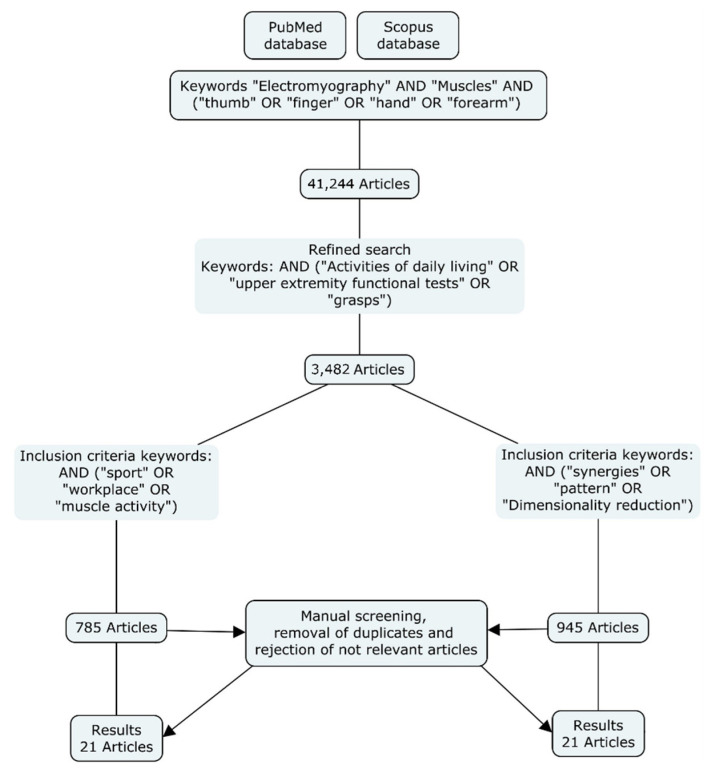
Methodology followed for the literature search.

## Data Availability

The study does not report any data.

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
