# Peer review of "A Systematic Review of EMG Applications for the Characterization of Forearm and Hand Muscle Activity during Activities of Daily Living: Results, Challenges, and Open Issues"

_sensors, 2021, doi:10.3390/s21093035_

Round 1

Reviewer 1 Report

Authors proposed a review on papers investigating activities of daily living performed with the forearm or hand, and studies investigating hand muscle synergies. The paper is clear but I think that its purpose should be more clarily stated. Why you investigated the only studies about activity of daily living? Why all papers collecting EMG from forearm and hand muscles could not be added in your review? Are not they simulating activity of daily living? Therefore, I think a clear definition of ‘activity of daily living’ is lacking. However, once this point is clarified, I recommend the publication of this study.

Following are some comments:

How do you define activities of daily living? E.g. is grasping always an ADL or is it an ADL only when it is finalized to a representative action (as it looks you suggest in line 216-217)? I think this definition should be stated in the introduction.

I suggest to add the day when you looked for the papers on Scopus and PubMed to justify the lack of some papers in your review, published after that day.

Line 36: The phrase looks to me not precise. I think it’s quite clear the role of different muscles of the hand (that is usually described by the muscle name, e.g. Extensor Pollicis brevis, Extensor indicis etc.). The problem is how can the Central Nervous System dare with the high redundancy to choose a specific muscle pattern.

Line 45: I suggest t0 add some studies made by the group leaded by Dario Farina about hand prosthesis, such as: Hahne et al., Longitudinal Case Study of Regression-Based Hand Prosthesis Control in Daily Life, Front. Neurosci., 2020

Line 67: another difference between surface EMG and needle EMG is the surface of the muscle on which the signal is acquired: bipolar surface EMG acquire the signal from, roughly, all the fibers between the two electrodes, while needle EMG acquire the signal from few fibers

Line 88: also surface EMG, collected with high density EMG grids, allows the identification of motor unit firing rates and motor unit action potentials, but only for motor units on the surface of the muscle (A. Holobar et al., “Estimating motor unit discharge patterns from high-density surface electromyogram,” Clin. Neurophysiol., vol. 120, pp. 551–562, 2009)

Line 234: I suggest to cite the work of Santello (e.g. Santello et al. 1998, Postural hand synergies for tool use, or Santello et al. 2013, Neural bases of hand synergies), who identified kinematic hand synergies, i.e. some hand postures whose combination, such as muscle synergies, explain most of the hand postures. I think these studies are strongly related with your review, even if authors did not used EMG, and needs to be cited, even if you choose not to add them in table 2

Line 309: Although Gazzoni and collaborators used an approach that was similar to the one used to identify synergies, the purpose of their study was not the identification of the muscle synergies, and they did not cite muscle synergies, but the description of an algorithm to discern different muscles from high density EMG. While I think this study is relevant for your review, and I agree that you added it, I suggest to add also the study by Tanzarella et al. 2020, Non-invasive analysis of motor neurons controlling the intrinsic and extrinsic muscles of the hand.

Reference 58 looks not appropriate for your review because no muscles acting on the wrist or hand are recorded. I’m afraid that authors confused the Pronator Teres muscle (that they affirmed it was collected by D’Andola and collaborators), with Teres Major (the muscle acting on the shoulder that D’Andola actually collected).

Some papers are missing (depending on the date in which you performed your review, see my previous comment), such as:

Gracia-Ibáñez et al., 2020, Sharing of hand kinematic synergies across subjects in daily living activities, Scientific Reports 

Jarrassé et al., 2014, Analysis of hand synergies in healthy subjects during bimanual manipulation of various objects,  Journal of NeuroEngineering and Rehabilitation

Author Response

Authors proposed a review on papers investigating activities of daily living performed with the forearm or hand, and studies investigating hand muscle synergies. The paper is clear but I think that its purpose should be more clarily stated. Why you investigated the only studies about activity of daily living? Why all papers collecting EMG from forearm and hand muscles could not be added in your review? Are not they simulating activity of daily living? Therefore, I think a clear definition of ‘activity of daily living’ is lacking. However, once this point is clarified, I recommend the publication of this study.

RESPONSE: Many thanks to the reviewer for his/her comments. As the reviewer points out, the review focused on activities of daily living, referred to as those elementary tasks that allow anyone to function with minimal autonomy and independence. Carrying out these activities is essential to ensure a full and autonomous life, that is why we focused our study on these activities. Thus, as suggested by the reviewer, a clear definition of activity of daily living has been added so that we believe that the purpose of the paper is now clearer. The other comments made by the reviewer have been also addressed.

Following are some comments:

How do you define activities of daily living? E.g. is grasping always an ADL or is it an ADL only when it is finalized to a representative action (as it looks you suggest in line 216-217)? I think this definition should be stated in the introduction.

I suggest to add the day when you looked for the papers on Scopus and PubMed to justify the lack of some papers in your review, published after that day.

RESPONSE: Definition of ADL has been added in the Introduction section: “ADLs refers to those elementary tasks that allow anyone to function with minimal autonomy and independence, including any daily activity that we perform for self-care, work, housework and leisure”. Furthermore, the month and year until we looked for the papers have been added.

Line 36: The phrase looks to me not precise. I think it’s quite clear the role of different muscles of the hand (that is usually described by the muscle name, e.g. Extensor Pollicis brevis, Extensor indicis etc.). The problem is how can the Central Nervous System dare with the high redundancy to choose a specific muscle pattern.

RESPONSE: We agree. The phrase has been rephrased to: “However, both the specific role of the different muscles in ADL and how the Central Nervous System dares with this redundant and complex muscular system are still unknown [9]

Line 45: I suggest t0 add some studies made by the group leaded by Dario Farina about hand prosthesis, such as: Hahne et al., Longitudinal Case Study of Regression-Based Hand Prosthesis Control in Daily Life, Front. Neurosci., 2020

RESPONSE: In the Introduction section, we have added some studies made by the group of Dario Farina about hand prostheses.

Line 67: another difference between surface EMG and needle EMG is the surface of the muscle on which the signal is acquired: bipolar surface EMG acquire the signal from, roughly, all the fibers between the two electrodes, while needle EMG acquire the signal from few fibers

RESPONSE: This information has been added in the new version of the manuscript: “Surface electrodes are placed on the skin directly over the muscles, recording the signal from all the fibers under the two electrodes. Intramuscular electrodes can be indwelling (also known as needle) or fine wire electrodes (Fw-EMG), and are inserted through the skin directly into the muscle [20], thus recording the signals from only few fibers”.

Line 88: also surface EMG, collected with high density EMG grids, allows the identification of motor unit firing rates and motor unit action potentials, but only for motor units on the surface of the muscle (A. Holobar et al., “Estimating motor unit discharge patterns from high-density surface electromyogram,” Clin. Neurophysiol., vol. 120, pp. 551–562, 2009)

RESPONSE: Thanks for this suggestion. This alternative for identify motor units has been added in the new version of the manuscript: “High density EMG grids allow also the identification of the MUAPs, but only for motor units on the surface of the muscle [22]

Line 234: I suggest to cite the work of Santello (e.g. Santello et al. 1998, Postural hand synergies for tool use, or Santello et al. 2013, Neural bases of hand synergies), who identified kinematic hand synergies, i.e. some hand postures whose combination, such as muscle synergies, explain most of the hand postures. I think these studies are strongly related with your review, even if authors did not used EMG, and needs to be cited, even if you choose not to add them in table 2

RESPONSE: We agree with the reviewer. Some of the works suggested were already cited in the manuscript. Now, in the Introduction section, we have added some additional works as Santello et al. 1998, Postural hand synergies for tool use, along with the works suggested in the reviewer’s last comment.

Line 309: Although Gazzoni and collaborators used an approach that was similar to the one used to identify synergies, the purpose of their study was not the identification of the muscle synergies, and they did not cite muscle synergies, but the description of an algorithm to discern different muscles from high density EMG. While I think this study is relevant for your review, and I agree that you added it, I suggest to add also the study by Tanzarella et al. 2020, Non-invasive analysis of motor neurons controlling the intrinsic and extrinsic muscles of the hand.

RESPONSE: We agree with the reviewer. As he/she commented, Gazzoni and collaborators’ work was added not because of its relevance in the identification of muscular synergies, but because of their method to identify areas of sEMG in the forearm with similar muscular activation. Now, the work suggested has been also added: “Tanzarella et al [107] performed isometric contractions with each finger and with three combinations of fingers in opposition with the thumb. They observed a low dimensional control of motor neurons across multiple intrinsic and extrinsic muscles.

Reference 58 looks not appropriate for your review because no muscles acting on the wrist or hand are recorded. I’m afraid that authors confused the Pronator Teres muscle (that they affirmed it was collected by D’Andola and collaborators), with Teres Major (the muscle acting on the shoulder that D’Andola actually collected).

RESPONSE: We agree with the reviewer. The reference D’Andola and collaborators (A Spatiotemporal characteristics of muscle patterns for ball catching. Front. Comput. Neurosci. 2013, 7, 107, doi:10.3389/fncom.2013.00107) was deleted from Table 2. However, it has been cited in the text, as it provides useful information on muscle synergies.

Some papers are missing (depending on the date in which you performed your review, see my previous comment), such as:

Gracia-Ibáñez et al., 2020, Sharing of hand kinematic synergies across subjects in daily living activities, Scientific Reports

Jarrassé et al., 2014, Analysis of hand synergies in healthy subjects during bimanual manipulation of various objects,  Journal of NeuroEngineering and Rehabilitation.

RESPONSE: The works suggested have been added in the Introduction section, as an evidence of the existence of synergies.

.

Reviewer 2 Report

L13

Insert electromyography (EMG)

L15

“of the hand.” Consider inserting the forearm too.

L21

Please, a think that “improved” could be better in present simple “improve”.

L45

What did you mean with “cumbersome”? Could you improve the sentence, please?

L143

“…long extensor and flexor muscles of the thumb (EPL, FPL)” you have shown in the abbreviation “Extensor Policis Longus”, maybe it is better to keep the same name in both.

L198-200

“A wide variety of clinical tests (such as the Jebsen Taylor Hand Function (JTHF) test [71], Chedoke Arm and Hand Activity Inventory (CAHAI) [72], or SHFT [73] are often used to evaluate and track functional recovery of the upper extremity simulating ADL.”

You opened the first parenthesis, but I think you left it open in the sentence. I think it closes after SHFT [73].

P16 Table 2

Ricci et al. [62], why the upper case in the R in the sentence “In the Reaching phase, the main movements observed were shoulder flexion and…”?

Author Response

RESPONSE: We thank the reviewer for his/her comments and suggestions. We have addressed all his/her comments.

L13

Insert electromyography (EMG)

RESPONSE: We have inserted it.

L15

“of the hand.” Consider inserting the forearm too.

RESPONSE: We agree. We have inserted ‘forearm’ too.

L21

Please, a think that “improved” could be better in present simple “improve”.

RESPONSE: We agree, and we have corrected it.

L45

What did you mean with “cumbersome”? Could you improve the sentence, please?

RESPONSE: We have changed the word “cumbersome” by the word “complex”. Now, the sentence is clearer.

L143

“…long extensor and flexor muscles of the thumb (EPL, FPL)” you have shown in the abbreviation “Extensor Policis Longus”, maybe it is better to keep the same name in both.

RESPONSE: We agree with the reviewer that it is better to keep the same name in both. We have changed the terminology used within the manuscript to:” Extensor Policis Longus and Flexor Policis Longus (EPL, FPL)

L198-200

“A wide variety of clinical tests (such as the Jebsen Taylor Hand Function (JTHF) test [71], Chedoke Arm and Hand Activity Inventory (CAHAI) [72], or SHFT [73] are often used to evaluate and track functional recovery of the upper extremity simulating ADL.”

You opened the first parenthesis, but I think you left it open in the sentence. I think it closes after SHFT [73].

 RESPONSE: We have closed the parenthesis after SHFT [73].

P16 Table 2

Ricci et al. [62], why the upper case in the R in the sentence “In the Reaching phase, the main movements observed were shoulder flexion and…”?

RESPONSE: The upper case “R” of reaching has been changed to a lower case.

Round 2

Reviewer 1 Report

The authors modified the manuscript according to my suggestions and I suggest the publication of the study.

I suggest only a minor change:

line78:

but only for motor units on the surface of the muscle -> but only for those motor units whose innervation zone was close to the surface of the muscle 

Author Response

The authors modified the manuscript according to my suggestions and I suggest the publication of the study.

I suggest only a minor change:

line78:

but only for motor units on the surface of the muscle -> but only for those motor units whose innervation zone was close to the surface of the muscle 

Response: Many thanks to the reviewer for his/her suggestion. We have changed the phrase.